Estimated projection of oral squamous cell carcinoma annual incidence from twenty years registry data: a retrospective cross-sectional study in Indonesia

http://orcid.org/0000-0003-2342-8133 Rahadiani Nur 1 nur.rahadiani@ui.ac.id
http://orcid.org/0000-0001-6372-8240 Habiburrahman Muhammad 2
Stephanie Marini 1
Handjari Diah Rini 1
Krisnuhoni Ening 1
1 Department of Anatomical Pathology, Faculty of Medicine Universitas Indonesia/Dr. Cipto Mangunkusumo Hospital , Jakarta , Indonesia
2 Faculty of Medicine Universitas Indonesia/Dr. Cipto Mangunkusumo Hospital , Jakarta , Indonesia
Albertini Maria Cristina
Electronic publication date: 2023 Aug 29
Publication date: 2023
Volume: 11
Electronic Location ID: e15911
Received 2022 Dec 22; Accepted 2023 Jul 26
Copyright: © 2023 Rahadiani et al.
Copyright year: 2023
Copyright holder: Rahadiani et al.
License: This is an open access article distributed under the terms of the Creative Commons Attribution License, which permits unrestricted use, distribution, reproduction and adaptation in any medium and for any purpose provided that it is properly attributed. For attribution, the original author(s), title, publication source (PeerJ) and either DOI or URL of the article must be cited.
License URL: https://creativecommons.org/licenses/by/4.0/

Keywords: Incidence, Trends, Forecasting, Oral squamous cell carcinoma, Indonesia

Funding: Ministry of Research and Technology/National Agency for Research and Innovation Research and Community Service Information System (SIMLITABMAS) Top Basic Research in University (PDUPT) NKB-122 This study was part of a research project supported by the Ministry of Research and Technology/National Agency for Research and Innovation through the Research and Community Service Information System (SIMLITABMAS) and Top Basic Research in University (PDUPT) grant scheme [grant number NKB-122, year 2021]. The funders had no role in study design, data collection and analysis, decision to publish, or preparation of the manuscript.

==============================
Background

The incidence of oral squamous cell carcinoma (OSCC) has not been well documented in Indonesia. Thus, we aimed to analyze trends and clinicopathological profiles of OSCC cases in Indonesia, focusing on differences between age and sex groups.

Methods

A cross-sectional study was conducted in Indonesia’s main referral hospital, analyzing 1,093 registered OSCC cases from 2001 to 2020. Trend analysis was performed using Joinpoint regression analysis to determine the annual percentage change (APC) for overall cases and each case group based on age, sex, and anatomical subsites. APC significance was assessed using a Monte Carlo permutation test. The projection of case numbers for the following 5 years (2021–2025) was estimated using linear/non-linear regression analysis and presented as a mathematical function. The significance of the trend slope was measured using an ANOVA test. Demographic and clinicopathological characteristics of OSCC were analyzed according to age and sex, and their comparative analysis was assessed using Chi-square and its alternatives.

Results

The incidence of OSCC in female patients and in the tongue and buccal mucosa showed a positive trend (APC 2.06%; 3.48%; 8.62%, respectively). Moreover, the incidence of OSCC overall, and in women with OSCC, is projected to increase significantly in the next 5 years following the quadratic model. The mean age of patients was 51.09 ± 14.36 years, with male patients being younger than female patients. The male-to-female ratio was 1.15, and 36.5% of these patients were categorized as young (≤45 years old). The tongue was the predominantly affected site. Prominent pathologic characteristics included well-differentiation, keratinization, and grade I of Bryne’s (1992) cellular differentiation stage. Most patients presented with advanced staging, lymphovascular invasion, and uninvaded margins. Tumor sites and staging varied according to age, while age and tumor sites differed between sexes.

Conclusion

The rising incidence trends of OSCC among Indonesian patients, both in the past and projected future, are concerning and warrant attention. Further research into risk factors should be conducted as preventive measures.

Introduction

The case number, morbidity, and mortality of oral cancer are rapidly increasing on a global scale (Sun et al., 2023). Squamous cell carcinoma is most prevalent among oral cancers, accounting for 90% of cases, and carries a poor prognosis, with a 5-year survival rate of 53.3–62.7% (Bai, Zhang & Wei, 2020). Oral squamous cell carcinoma (OSCC) can affect any part of the oral cavity, but the tongue and floor of mouth (FOM) are the most commonly affected (Farhood et al., 2019). OSCC occurs more frequently in Southeast Asia, including Indonesia (Johnson, Jayasekara & Amarasinghe, 2011). In 2020, oral cancer was ranked 17th among all malignancies in Indonesia, with 5,780 reported cases and 3,087 deaths (International Agency for Research on Cancer, 2021).

The incidence of OSCC has recently been increasing among youth worldwide, emerging as a significant public health issue, accounting for 4–13% of all OSCC cases (Santos et al., 2016). The prognosis of OSCC in young adults remains controversial, with one study indicating that younger patients experience poorer outcomes and exhibit more aggressive cellular behavior than their older counterparts (Sharma & Singh, 2016), while another study found no difference in prognosis between both age groups (Sasaki et al., 2005). On the other hand, the increasing incidence of OSCC in females is also concerning, as previously reported (Patel et al., 2011). Several studies have suggested that OSCC may display different characteristics, risk factors, and prognoses based on gender (Lee et al., 2021). However, knowledge of distinct clinicopathological aspects in OSCC among these subpopulations in Indonesia is limited.

Epidemiology-based pathology studies on OSCC cases in Indonesia are scarce. A barrier in extrapolating data of OSCC cases and comparing pathology profiles across the Indonesian population arises from differing diagnostic criteria, classifications, and utilization of histopathology services. Therefore, this study aimed to conduct a comprehensive investigation and measure the incidence rate of OSCC in Indonesia’s national tertiary hospital between 2001 and 2020, focusing on young and female patients. The study also highlighted notable demographic, clinicopathological, and histological findings in young and female patients with OSCC. Additionally, few previous studies have attempted to predict the growth of OSCC incidence. One recent study conducted in Spain made an effort to forecast the deaths caused by oral cavity and oropharyngeal cancer in 2044 (Infante-Cossio et al., 2022). Thus, our aim was also to depict the annual incidence trends and provide an estimated projection of OSCC cases for the next 5 years (2021–2025). This information may serve as a foundation for future research, development of diagnostic and screening methods, treatment strategies, surveillance measures, educational plans, resource allocation, and effective prevention approaches to reduce OSCC incidence and mortality.

Materials and Methods

Study design, cancer incidence data, and eligibility criteria

This retrospective cross-sectional study examined 1,093 OSCC cases treated at Dr. Cipto Mangunkusumo Hospital from January 2001 to December 2020. Our institution is the largest national referral hospital in Indonesia, handling 18,216 cancer cases from 2008 to 2012, as documented in the hospital’s cancer registry data (Gondhowiardjo et al., 2021). All patients with OSCC who underwent histopathological examinations were included in our study sample. Exclusion criteria included missing slides, changes of diagnosis following reassessment, and duplicate inputs due to multiple specimen-taking procedures performed on the same patient; all sample-collecting processes are described in Fig. 1. Data from tissue blocks stained with hematoxylin and eosin (H&E), slide archives, and the patient’s clinical profiles were analyzed to obtain clinicopathological information. The authors assessed the pathological diagnosis separately and held discussions to establish the final diagnosis.

Figure 1 Study flowchart.

The study flowchart for selecting and recruiting cases, followed by data analysis for the total sample and particular subgroups.

Study variables and parameters

This study analyzed demographic and clinicopathological profiles, including registration year, age, sex, and tumor subsites. The cut-off age for defining young patients in a study is arbitrary, and reaching a unanimous agreement on what age should be categorized as ‘young’ is challenging. In this study, ‘young patients’ were defined as those aged less than or equal to 45 years old, consistent with previous studies (Iamaroon et al., 2004; Santos et al., 2016). The International Classification of Diseases 10th Edition (ICD-10) coding system was used to recognize 15 tumor subsites. The study measured the following histopathological parameters: (1) keratinization, based on the World Health Organization (WHO) and ICD-10 code groupings, (2) WHO histological differentiation classified as well-, moderate-, and poor differentiation, and (3) the degree of cellular differentiation according to the Bryne et al. (1992) grading system, categorized as 4–8 (Grade I), 9–12 (Grade II), and 13–16 (Grade III) (Wagner et al., 2017). Histological differentiation records were unavailable for 382 patients, and cellular differentiation records were unavailable for 526 patients. Surgical resection samples were examined to assess surgical margins, tumor presence, and lymphovascular invasion (LVI). According to a previous study, surgical margins were considered positive if the tumor was present at the edge (1 mm) or close to (1–5 mm) healthy tissue, and negative if the tumor was present 5 mm or further from the margins (Helliwell & Giles, 2016). Clinical TNM staging was applied to patients who underwent surgical treatments following the eighth edition American Joint Committee on Cancer (AJCC) staging manual (Kowalski & Köhler, 2019). The classification of T, N, and M was simplified in further statistical analysis. Clinical staging was also categorized based on the degree of disease progression (early-stage/I–II and advanced-stage/III–IV). Staging information was not available for 38 patients.

Statistical analysis

Data were analyzed using descriptive and inferential statistics with the Statistical Package for Social Sciences (SPSS) version 25.0 (SPSS Inc., Chicago, IL, USA). Chi-Square and its alternative tests, including Fisher’s exact, Mann-Whitney U, and Kruskal-Wallis tests were utilized. The probability value was set to 0.05. Missing data were excluded before data analysis (Dahlan, 2017).

Trend analysis

Based on historical data, cancer trends can be predicted using appropriate mathematical methods. This study employed two approaches to assess OSCC trends: Joinpoint regression analysis for past data and best-fitted regression models for forecasting future incidence. Joinpoint regression (utilizing the Joinpoint package 4.9.1.0; https://surveillance.cancer.gov/joinpoint/) was used to make inferences about trends over time based on the available past data (National Cancer Institute, 2022a). The widely used joinpoint regression method (Kim et al., 2000) connects multiple line segments on a logarithmic scale to represent cancer trends over time. It analyzes temporal trends in specific quantities like proportions, rates, or counts, which is valuable for cancer-related studies on incidence and mortality, offering quantitative insights into epidemiological research (Rea et al., 2017; National Cancer Institute, 2021). Trend analysis of OSCC through 20 years was presented as annual percent change (APC) with 95% confidence intervals (95% CI) (Liu et al., 2022). A Monte Carlo permutation test was automatically applied to the data to determine the significance of changes. This test selects the most appropriate line segment(s) in the time series data, with statistical significance denoted as p < 0.05 (Liu et al., 2022).

In performing the Joinpoint regression analysis, we considered parameters such as the optimizing method (grid search), maximum join point, and closeness of different joinpoints. The software automatically employs the grid search method, which uses permutation tests to find the optimal number of joinpoints. It detects joinpoints in trends through a numerical search process and fits a linear regression between consecutive joinpoints (Kafle, 2014). Subsequently, based on algorithmic recommendations, the default maximum number of joinpoints was determined based on the number of data points: 0 joinpoints for 0–6 data points, no joinpoints for 7–11 data points, two joinpoints for 12–16 data points, three joinpoints for 17–21 data points, four joinpoints for 22–26 data points, and five joinpoints for ≥27 data points. For our 20-year time series data, the algorithm initially suggested three joinpoints. However, after evaluating all choices (0, 1, 2, and 3 joinpoints), we selected the best-fitted model with zero joinpoints, as it exhibited the most significant average APC and a more homogeneous interval, ensuring a more reliable time series generation. The parameter of joinpoint closeness was considered to assess the similarity of APCs between joinpoints. Close joinpoints have similar trends and prevent the inclusion of spurious joinpoints (Irimata et al., 2022; National Cancer Institute, 2022b). Initially, we intended to use Hudson’s method, but it has been disabled in the recent software version due to algorithmic issues in setting the closeness of each joinpoint and at the endpoint of the data. Consequently, we opted for the grid search method as an alternative, which uses data points as the grid and is faster when using the permutation test for model selection compared to Hudson’s method (Irimata et al., 2022).

Estimating projection model

In this study, we also aimed to project or forecast future cancer incidence rates using best-fitted data in various regression models, including linear, quadratic, exponential, and S-shaped curves, with the help of statistical software (Minitab® 19.1 (64-bit)). Model equations predicting OSCC cases can be visualized as various mathematical functions (Claudio, Miller & Huggins, 2014; Musa, 2017; Ashoor, Kazem & Gore, 2021; Wah et al., 2021). Each function has its algebraic form involving variables and coefficients. The linear function is denoted as Yt = b0 + (b1 * t), the quadratic function as Yt = b0 + b1 * t + (b2 * t2), the exponential function as Yt = b0 + (b1t), and the S-curve function as Yt = (10a)/(b0 + b1 * b2t). In these equations, Yt represents the variable of interest, a and b0 as a constant, b1 and b2 as coefficients, and t as the time unit value. The Minitab software automatically generated the accuracy measurement (MAPE, mean absolute percent error; MAD, mean absolute deviation; and MSD, mean squared deviation or multiple seasonal decompositions) for each curve (Kasapoalu, 2016; Minitab Inc, 2021).

MAPE is a measure of forecasting accuracy that compares the forecasting error to actual data values. However, MAPE may not be suitable when the data contains extremes or near-zero values, as it can result in distorted error rates. To assess accuracy, it is essential to consider other measurements like MAD and MSD, in addition to MAPE. MAPE calculates the average absolute percent error between forecasts and actuals, while MAD calculates the average absolute error regardless of the direction of the error. MSD considers the average of squared data deviations and is more sensitive to significant forecast errors. Considering both MAD and MSD can help determine the best estimation method with the lowest deviation (Putri, Sukiyono & Sumartono, 2019; Wahyu, Rahmawati & Umam, 2022). These parameters have been employed in previous cancer research studies involving time series data (Earnest et al., 2019; Rahadiani et al., 2022a; Tudor, 2022). The best-fitted curve to be chosen as a model for subsequent projection is the one that has the lowest value for three parameters, or at least for two parameters out of three, or at least having the lowest value for MAPE (Tofallis, 2015; Musa, 2017; Minitab Inc, 2021).

Validating the forecasting model

In our study, we conducted rigorous validation to assess the predictive performance of our model. After evaluating different model types to capture the underlying trend, we found that the quadratic model provided the best fit. As detailed in Data S3, we then conducted a residual analysis to validate the trend. We examined the deviations between the fitted/predicted line and observed values. Residual plots helped identify non-linear relationships and outliers. We checked for normal distribution and constant variance of residuals, which indicate a good fit (Altman & Krzywinski, 2016; Elsayir, 2019). The R-squared test was used to assess the adequacy of the trend formula in fitting the data. It quantifies the proportion of variance explained by the model. However, achieving a 100% value is ideal but unattainable. Interpretation should consider additional metrics and practical significance (Chicco, Warrens & Jurman, 2021; Kang, Sharma & Marshall, 2021). Homoscedasticity was assessed by examining the plot of fitted values vs. residuals. Consistent variance across the x-axis indicated adherence to the assumption, while a discernible pattern suggested heteroscedasticity (Da Silva, Emídio & De Marchi, 2015; Newcastle University, 2019). Autocorrelation analysis was performed to assess the association between current and previous values. It validated assumptions and identified violations. A correlation close to zero indicated accurate forecasts, while deviations suggested incomplete capture of patterns or dependencies (Box & Pierce, 1970; Minitab Inc, 2023). Lastly, in assessing the significance of the slope, the ANOVA test for the regression model in SPSS was done, indicating a statistically significant trend if the p-value < 0.05 (Dahlan, 2017; Minitab Inc, 2021).

Ethical approval

The Ethics Committee of the Faculty of Medicine Universitas Indonesia and Dr. Cipto Mangunkusumo Hospital have granted ethical approval for this research (protocol number 21-03-0246) under the decision number KET-178/UN2.F1/ETIK/PPM.00.02/2021. All methods and study results were reviewed and reported following the STROBE guidelines (Supplementary Data S1) for cross-sectional studies (Vandenbroucke et al., 2014). Based on our board’s ethical research policy (No: ND-826/UN2.F1/ETIK/PPM.00.02/2022), direct informed consent has been waived for studies using existing data collections, documents, pathological specimens, or other diagnostic specimens.

Results

Demographic characteristics

Demographic characteristics of patients presented in Fig. 2A shows that the majority of OSCC patients were elderly. However, importantly, more than a third of the overall patients were young. The growth of OSCC cases in both young and old patients was primarily observed in the last 5-year period (2016–2020). Compared to the first, second, and third 5-year periods, the growth rates of OSCC cases were 7.6%, 6.9%, and 8.3%, respectively.

Figure 2 Demographic characteristics and incidence trends of 1,093 OSCC cases from 2001–2020.

(A) The distribution of OSCC cases per 5 years according to two age groups (young, ≤45 years vs. old, >45 years) and their increased number of total cases in four short periods. (B) The volatile changes of OSCC cases over 20 years among male and female patients. (C) The percentage of OSCC numbers by sex in four sequential 5-year periods. (D) The number of OSCC cases in different age categories (10-year age groups). (E) The proportion of OSCC cases among young and old patients and looked more detailed into their sex proportion.

As seen in Figs. 2B and 2C, the male population consistently outnumbered the female population, with a male-to-female ratio of 1.15. Although male cases exceeded female cases every year and every 5 years, there was an increase in female subjects, particularly in 2006–2010 and 2016–2020. Figure 2D demonstrated that the highest proportion of OSCC patients belonged to the age group of 51–60, followed by those 41–50 and 61–70 age groups. The mean age of the overall population was 51.09 ± 14.36 years old, 50.35 ± 14.45 years old in males, and 51.81 ± 14.27 years old in females.

Trend analysis

Past trends of OSCC cases over a 20-year were analyzed using Joinpoint regression analysis, and the APCs for each subgroup of OSCC cases are presented in Table 1 (detailed data provided in Data S2). From 2001 to 2020, the average occurrence of OSCC in our institution was 55 cases per year. OSCC presentation varied according to age, sex, and tumor sites. Although not statistically significant, the APC of overall OSCC cases and age-specific trends showed a modest increase from 2001 to 2020. However, in the quadratic regression model, the increase in overall OSCC cases was statistically significant (p = 0.046), while age-specific trends remained nonsignificant (p > 0.05). A notable difference in trends between sexes was observed, with a significant rise in APC among women (2.06%) but a stable trend among men (1.27%). When examining the anatomical sites of OSCC, there was a significant increase in trends for cancers of the tongue (3.48%) and buccal mucosa (8.62%). Conversely, the trend for cancers of the FOM showed a decline (–5.17%). Trends for cancer at other sites remained stable during the study period.

Table 1 Number, percentage, and incidence trends of OSCC cases from 2001 to 2020 (n = 1,093).

Characteristics	Cases	APC†	p-value	95% CI‡	Interpretation of changes	
n	%	
All cases	1,093	100	1.60	0.059	[−0.07 to 3.30]	Stable (Not significant)	
Age							
≤45 years (young)	399	36.5	1.95	0.132	[−0.64 to 4.61]	Stable (Not significant)	
>45 years (old)	694	63.5	1.46	0.087	[−0.23 to 3.17]	Stable (Not significant)	
Sex							
Males	585	53.5	1.27	0.207	[−0.76 to 3.35]	Stable (Not significant)	
Females	508	46.5	2.06	0.025	[0.29−3.86]	Increase (Significant)	
Tumor origin sites							
Tongue	702	64.2	3.48	0.001	[1.52−5.47]	Increase (Significant)	
Mouth NOS§	186	17.0	−2.20	0.152	[−5.22 to 0.90]	Stable (Not significant)	
Palate	78	7.1	−1.44	0.572	[−6.54 to 3.93]	Stable (Not significant)	
Gingiva	56	5.1	4.01	0.131	[−1.28 to 9.59]	Stable (Not significant)	
Lip	45	4.1	−3.33	0.233	[−8.64 to 2.28]	Stable (Not significant)	
Buccal mucosa	12	1.1	8.62	<0.001	[4.70−12.70]	Increase (Significant)	
FOM	14	1.3	−5.17	0.020	[−9.23 to −0.93]	Decrease (Significant)	
Notes:

† APC, Annual percent change (%).

‡ CI, Confidence interval.

§ NOS, Not otherwise specified.

The percentage ‘%’ values represent column percentages.

Bold styling indicates significance.

Estimated projection of future cases

Analysis of OSCC cases in Fig. 3 reveals an increasing trend over the past 20 years for all subjects and subpopulations (young/old and male/female patients) using quadratic regression analysis as the best-fit model curve over other curve forms. Further details on selecting the best-fit model curve are described in Data S3. In the regression analysis, a mathematical function was generated to estimate the projection of OSCC incidence in the next 5 years for overall group (Yt = 53.13 – 1.46t + 0.1174t2), the young subgroup (Yt = 18.66 – 0.49t + 0.0451t2), the older subgroup (Yt = 34.48 – 0.97t + 0.0723t2), males (Yt = 28.98 – 0.73t + 0.0554t2), and females (Yt = 24.16 – 0.729t + 0.0620t2). The trends of OSCC cases for all cases, young and old patients, as well as male and female patients, are expected to gradually increase over the next 5 years (2021–2025). The most significant changes in trend slopes were observed in the projections for the overall patient population (p = 0.046) and the female subpopulation (p = 0.021).

Figure 3 Trend analyses of 1,093 OSCC cases over 20 years and prediction of OSCC incidence in the next 5 years in Dr. Cipto Mangunkusumo Hospital using non-linear quadratic regression model analysis.

(A) All patients. (B) Young patients. (C) Old patients. (D) Male patients. (E) Female patients. An increasing trend occurred significantly for total and female cases (p < 0.05). Notes: Yt, variable; t, time; MAPE, mean absolute percent error; MAD, mean absolute deviation; MSD, mean square deviation; *, significant p-value obtained from ANOVA test for curve estimation.

The validation results of our forecasting models (quadratic models) for five cases (overall cases, young and old subgroups, and male and female groups) demonstrated their suitability as OSCC projection models. The normality tests for the residuals indicated a close alignment with the “zero line” or “identity line,” confirming a normal distribution (p > 0.05). The R-square values for all quadratic models were >0%, ranging from 14.4% to 34.1%, indicating a reasonable fit of the regression model to the data. The plot of fitted values vs. residuals exhibited consistent variance, satisfying the assumption of homoscedasticity. Additionally, the correlation between fitted and residual data for overall cases and groupings was near zero, indicating fixedness, independence, and the absence of systematic patterns. These validation results affirm the model’s ability to accurately capture underlying data patterns and provide accurate forecasts.

Sub-analysis of age-specific characteristics

Based on the examination of patient characteristics and clinicopathological parameters provided in Table 2, the majority of tumors exhibited keratinization (78.4%), were well-differentiated (53.0%), and had grade I cellular differentiation according to the Bryne et al. (1992) grading system (55.9%). The tongue was the most prevalent tumor site among all patients (64.2%). There was a significantly higher proportion of OSCC originating from the tongue and FOM in the young population compared to older people. OSCC originating from the mouth not otherwise specified (NOS), palate, gingiva, lip, and buccal mucosa were predominantly found in the more aging population (p < 0.001). There were no statistically significant differences in other demographic and clinicopathological factors between younger vs. older patients.

Table 2 Comparison of demographic characteristics and clinicopathological features between two age groups (n = 1,093).

Demographic characteristics and
clinicopathological features	Age groups	Total	p-value	
≤45 years
(n = 399)	>45 years
(n = 694)	
n	%	n	%	n	%	
Registration year							0.447*	
2001–2005	88	22.1	164	23.6	252	23.1		
2006–2010	101	25.3	159	22.9	260	23.8		
2011–2015	81	20.3	164	23.6	245	22.4		
2016–2020	129	32.3	207	29.8	336	30.7		
Sex							0.493*	
Male	219	54.9	366	52.7	585	53.5		
Female	180	45.1	328	47.3	508	46.5		
Tumor sites							<0.0001**	
Tongue	293	73.4	409	58.9	702	64.2		
Mouth NOS†	43	10.8	143	20.6	186	17.0		
Palate	26	6.5	52	7.5	78	7.1		
Gingiva	18	4.5	38	5.5	56	5.1		
Lip	10	2.5	35	5.0	45	4.1		
Buccal mucosa	3	0.8	9	1.3	12	1.1		
FOM‡	6	1.5	8	1.2	14	1.3		
Keratinization							0.974*	
Yes	312	78.2	545	78.5	857	78.4		
No	60	15.0	101	14.6	161	14.7		
Non-specific	27	6.8	48	6.9	75	6.9		
WHO histological differentiation							0.477*	
Well	143	53.6	234	52.7	377	53.0		
Moderate	67	25.1	126	28.4	193	27.1		
Poor	36	13.5	45	10.1	81	11.4		
Undifferentiated	21	7.9	39	8.8	60	8.4		
Missing data					382			
Bryne et al. (1992) of cellular differentiation							0.191*	
Grade I	108	51.4	209	58.5	317	55.9		
Grade II	75	35.7	115	32.2	190	33.5		
Grade III	27	12.9	33	9.3	60	10.6		
Missing data					526			
Specimen type							0.871*	
Biopsy	278	69.7	492	70.9	770	70.4		
Excision	5	1.3	7	1.0	12	1.1		
Resection	116	29.1	195	28.1	311	28.5		
Notes:

† NOS, Not otherwise specified.

‡ FOM, Floor of the mouth.

* Chi-Square.

** Kruskal-Wallis.

The percentage ‘%’ values represent column percentages.

Values in bold indicate a statistically significant association between the variables.

Upon examining the sub-analysis of 311 patients who underwent resection, as shown in Table 3, it was observed that tumors were typically extensive (T4: 60.4%), did not involve lymph nodes (N0: 43.6%), and did not exhibit distant metastasis (M0: 98.5%). The majority of patients presented at an advanced stage of the disease (III–IV: 81.3%). The surgical outcomes were generally satisfactory, with an 85.9% rate of resection margins being tumor-free. However, LVI was present in 52.7% of cases. Furthermore, our findings indicate a difference in the characteristics of OSCC between young and elderly individuals. Young patients tended to present at a more advanced stage (p < 0.05) than older patients.

Table 3 Comparison of clinicopathological features between age groups among patients who underwent resection (n = 311).

Clinicopathological features	Age groups	Total	p-value	
≤45 years
(n = 116)	>45 years
(n = 195)	
n	%	n	%	n	%	
Tumor size							0.253*	
T1	6	5.5	10	6.1	16	5.9		
T2	15	13.8	37	22.6	52	19.0		
T3	15	13.8	25	15.2	40	14.7		
T4	73	67.0	92	56.1	165	60.4		
Missing data					38			
Node involvement							0.505*	
N0	45	41.3	74	45.1	119	43.6		
N1	29	26.6	48	29.3	77	28.2		
N2	35	32.1	42	25.6	77	28.2		
Missing data					38			
Distant metastasis							0.679**	
M0	107	98.2	162	98.8	269	98.5		
M1	2	1.8	2	1.2	4	1.5		
Missing data					38			
Staging							0.252***	
I	3	2.8	10	6.1	13	4.8		
II	11	10.1	27	16.5	38	13.9		
III	16	14.7	26	15.9	42	15.4		
IVA	62	56.9	87	53.0	149	54.6		
IVB	14	12.8	12	7.3	26	9.5		
IVC	3	2.8	2	1.2	5	1.8		
Missing data					38			
Staging group							0.044*	
I–II (early stage)	14	12.8	37	22.6	51	18.7		
III–IV (advanced stage)	95	87.2	127	77.4	222	81.3		
Missing data					38			
LVI †							0.171*	
Negative	49	42.2	98	50.3	147	47.3		
Positive	67	57.8	97	49.7	164	52.7		
Margin of resection							0.593*	
Negative	98	84.5	169	86.7	267	85.9		
Positive	18	15.5	26	13.3	44	14.1		
Notes:

† LVI, Lymphovascular invasion.

* Chi-Square.

** Mann-Whitney.

*** Kruskal-Wallis.

The percentage ‘%’ values represent column percentages.

Values in bold indicate a statistically significant association between the variables.

Sub-analysis of sex-specific characteristics

As shown in Table 4, the proportion of female patients exceeded that of male patients in most age groups (11–20, 21–30, 31–40, 51–60, 71–80, and >80). Women also had a higher proportion of tumors originating in the tongue (68.3%), lip (5.3%), and buccal mucosa (1.8%). On the other hand, other clinicopathological factors described in Table 5 were not significantly different between male and female patients who underwent resection.

Table 4 Comparison of demographic characteristics and clinicopathological features between sexes (n = 1,093).

Demographics and
clinicopathological features	Sex	Total	p-value	
Male
(n = 585)	Female
(n = 508)	
n	%	n	%	n	%	
Age							0.029*	
11–20	4	0.7	5	1.0	9	0.8		
21–30	45	7.7	45	8.9	90	8.2		
31–40	93	15.9	85	16.7	178	16.3		
41–50	152	26.0	99	19.5	251	23.0		
51–60	139	23.8	152	29.9	291	26.6		
61–70	118	20.2	78	15.4	196	17.9		
71–80	25	4.3	36	7.1	61	5.6		
>80	9	1.5	8	1.6	17	1.6		
Tumor sites							<0.0001**	
Tongue	355	60.7	347	68.3	702	64.2		
Mouth NOS†	103	17.6	83	16.3	186	17.0		
Palate	59	10.1	19	3.7	78	7.1		
Gingiva	38	6.5	18	3.5	56	5.1		
Lip	18	3.1	26	5.3	45	4.1		
Buccal mucosa	3	0.5	9	1.8	12	1.1		
FOM‡	9	1.5	5	1.0	14	1.3		
Keratinization							0.798**	
Yes	463	79.1	394	77.6	857	78.4		
No	84	14.4	77	15.2	161	14.7		
Non-specific	38	6.5	37	7.3	75	6.9		
WHO histological differentiation							0.741**	
Well	204	53.7	173	52.3	377	53.0		
Moderate	105	27.6	88	26.6	193	27.1		
Poor	43	11.3	38	11.5	81	11.4		
Undifferentiated	28	7.4	32	9.7	60	8.4		
Missing data					382			
Bryne et al. (1992) of cellular differentiation							0.593**	
Grade I	168	57.1	149	54.6	317	55.9		
Grade II	93	31.6	97	35.5	190	33.5		
Grade III	33	11.2	27	9.9	60	10.6		
Missing data					526			
Notes:

† NOS, Not otherwise specified.

‡ FOM, Floor of the mouth.

* Kruskal-Wallis.

** Chi-Square.

The percentage ‘%’ values represent column percentages.

Values in bold indicate a statistically significant association between the variables.

Table 5 Comparison of clinicopathological features between sexes among patients who underwent resection (n = 311).

Clinicopathological features	Sex	Total	p-value	
Male
(n = 155)	Female
(n = 156)	
n	%	n	%	n	%	
Tumor size							0.901*	
T1	7	4.9	9	6.9	16	5.9		
T2	27	19.0	25	19.1	52	19.0		
T3	22	15.5	18	13.7	40	14.7		
T4	86	60.6	79	60.3	165	60.4		
Missing data					38			
Node involvement							0.353*	
N0	57	40.1	62	47.3	119	43.6		
N1	45	31.7	32	24.4	77	28.2		
N2	40	28.2	37	28.2	77	28.2		
Missing data					38			
Distant metastasis							0.679**	
M0	141	99.3	128	97.7	269	98.5		
M1	1	0.7	3	2.3	4	1.5		
Missing data					38			
Staging							0.989***	
I	6	4.2	7	5.3	13	4.8		
II	19	13.4	19	14.5	38	13.9		
III	23	16.2	19	14.5	42	15.4		
IVA	79	55.6	70	53.4	149	54.6		
IVB	13	9.2	13	9.9	26	9.5		
IVC	2	1.4	3	2.3	5	1.8		
Missing data					38			
Staging group							0.635*	
I–II (early stage)	25	17.6	26	19.8	51	18.7		
III–IV (advanced stage)	117	82.4	105	80.2	222	81.3		
Missing data					38			
LVI †							0.693*	
Negative	75	48.4	72	46.2	147	47.3		
Positive	80	51.6	84	53.8	164	52.7		
Margin of resection							0.340*	
Negative	136	87.7	131	84.0	267	85.9		
Positive	19	12.3	25	16.0	44	14.1		
Notes:

† LVI, Lymphovascular invasion.

* Chi-Square.

** Fisher’s Exact Test.

*** Kruskal-Wallis.

The percentage ‘%’ values represent column percentages.

Figure 4 provides histopathological findings of patients with OSCC who were examined at our institution. These findings are described in detail in Tables 2–5.

Figure 4 Histopathological features of OSCC. WHO histological differentiation.

(A) Welldifferentiated OSCC resembles typical squamous epithelium with abundant keratin pearls, H&E staining, M100X. (B) Moderately differentiated OSCC contains distinct nuclear pleomorphic and mitotic activity, such as bizarre mitoses and less keratinization, H&E staining, and M100X. (C) Poorly differentiated OSCC dominated by immature tumor cells, H&E staining, M100X. Bryne et al. (1992) score of cellular differentiation: (D) Pattern of the tumor with border infiltration, solid cords, bands, and/or strands, H&E staining, M100X. (E) Tumor cells with marked nuclear pleomorphism, H&E staining, M400X. (F) Lymphoplasmacytic infiltration among tumor mass, H&E staining, M100X. Features contributed to patients’ prognosis: (G) LVI, H&E staining, M100X. (H) Lymph node metastases showing tumor mass surrounded by lymphocyte rim of a lymph node, H&E staining, M40X. (I) Positive resection margins as evidenced by cancer cells at the edge (black ink) of the resection specimen, H&E staining, M100X.

Discussion

Epidemiologic studies on OSCC in Indonesia are limited. Previous documentation of OSCC incidence in Indonesia was sporadic, not comprehensive, limited in small sample sizes and study periods, and did not include trend analyses. Earlier Indonesian studies conducted in West Java (2014–2015) (Maulina et al., 2017), Yogyakarta (2011–2015) (Gracia et al., 2017), and Jakarta (2003–2013) (Purwanto et al., 2020) involved 95, 91, and 78 OSCC cases, respectively. In comparison, our present report has a relatively larger sample size, a longer study period, and a more comprehensive description of clinicopathological characteristics. However, the study’s generalizability and external validity may be limited as it was conducted in a single institution.

Our study identified an average of 55 OSCC cases per year, which is consistent with findings from other developing Asian countries, such as India (2007–2014: 301 cases) (Smitha, Mohan & Hemavathy, 2017), Iran (2006–2015: 587 cases) (Ghafari, Naderi & Razavi, 2019), and Pakistan (2013–2016: 115 cases) (Mahmood et al., 2018). The median age of our patients was 51 years old (range: 17–99), and most cases were in the 51–60 age group (Fig. 2D). Other studies have also shown that OSCC is most commonly diagnosed in individuals in their fifties to seventies (Johnson, Jayasekara & Amarasinghe, 2011). Notably, young patients accounted for a significant proportion (36.5%) of our study population, higher than the global average of 0.4–13% (Santos et al., 2016; Sharma & Singh, 2016). However, our findings are consistent with a study in Kuwait, where 35.8% of their OSCC cases were patients under 40 years old (Morris et al., 2000).

Furthermore, our analysis using forecasting models estimated a significant increase in OSCC cases over the next 5 years, particularly among female patients and overall subjects, based on quadratic models. Global Cancer Incidence, Mortality, and Prevalence (GLOBOCAN) data also predicts a continued rise in oral cancer incidence after 2020 in all countries in Southeast Asia, including Indonesia, with Brunei Darussalam and Cambodia are projected to have the steepest increases (Cheong et al., 2017; Chher et al., 2018; International Agency for Research on Cancer, 2021). However, Interpreting the quadratic models for forecasting OSCC cases in the next 5 years requires caution. The choice of quadratic models may lead to overfitting or underfitting, affecting prediction accuracy. Forecasting is inherently complex and uncertain due to multiple influencing factors influencing future circumstances. Underfitting occurs when a simple model fails to capture real-world complexity, while overfitting results from overly complex models with low bias and high variance (Lever, Krzywinski & Altman, 2016).

While we recognize that the quadratic model may capture noise or random fluctuations in the data, which can affect the reliability of predictions, previous studies suggest that this second-order polynomial model can be considered an excellent alternative to the best model (a third-order polynomial with customarily distributed noise) (Lever, Krzywinski & Altman, 2016). Data S3 also demonstrates that the quadratic model outperforms other available models based on accuracy and error measurements (MAPE, MAD, and MSD) in most cases (Lever, Krzywinski & Altman, 2016; Zhang & Jiang, 2021). This finding is consistent with other medical studies, where the quadratic model has been discovered to be superior for trend analysis and forecasting (Sharma & Nigam, 2020; Zhang & Jiang, 2021; Saitta et al., 2023). Therefore, the quadratic function is considered appropriate for describing the trend of OSCC cases. Additionally, a quadratic pattern signifies a non-linear trend in the data, and the ratio between the linear and quadratic terms falling within a meaningful range supports a non-monotonic trend (Moffat & Akpan, 2014).

Trend analysis according to age

In this study, the trend analysis of OSCC cases according to age, using the Joinpoint regression approach, revealed an insignificant increase in annual cases among both old and young patients, as shown in Table 1 and detailed in Data S2. The lack of case growth among older patients in our study is consistent with the phenomenon observed in a previous study. This finding is supported by a stable trend observed in US patients over 40 years old with tongue cancer from 1973 to 1997 (Schantz & Yu, 2002). In contrast to our findings, the Netherlands study from 1991 to 2010 reported an increasing trend in older Dutch patients (46–60 years old) during half of the study period, with a declining trend in the remaining half. However, similar to our results, a stable trend was observed in patients aged 45 years old or younger (van Dijk et al., 2016). In China during 1990–2017, the incidence of OSCC also showed different patterns, increasing for individuals aged 30 years and older, peaking at 65–69 years, but decreasing for individuals aged 29 years and younger (Yang et al., 2021). These discrepancies may be due to differences in age categorization used, which could affect incidence rate estimation and contribute to the contrasting results.

In both young and old patients, oral cancer may share similar classical risk factors, such as smoking and alcohol consumption (Llewellyn et al., 2004). Despite this, differences in prognosis and biological behavior exist between the two groups (Sarode et al., 2021). Younger patients may experience poorer outcomes and exhibit more aggressive cellular behavior (Mesquita et al., 2014). To test and validate the hypothesis, further research is needed to investigate time-dependent etiological factors and their duration of exposure associated with oral cancer incidence in both young and old patients.

Trend analysis according to sex

Men exhibited a higher prevalence of OSCC (53.5% male) (Figs. 2B and 2C), consistent with observations from other countries, including Taiwan (93.4%) (Lee et al., 2021), Thailand (57.4%) (Iamaroon et al., 2004), Iran (59.6%) (Ghafari, Naderi & Razavi, 2019), and Pakistan (71.3%) (Mahmood et al., 2018). Despite a higher number of men (Figs. 2B and 2C), we observed a temporary increase in OSCC cases among women during two consecutive periods: 2006–2010 and 2016–2020. The incidence trend among women significantly increased from 2001 to 2020, with an APC of 2.06% (95% CI [0.29–3.86]). Projected estimates suggest this trend will continue gradually over the next 5 years (Table 1 and Fig. 3E). In contrast, OSCC occurrence among males exhibited a more stable trend over 20 years and is projected to remain stagnant in the future. Our findings align with a study conducted in the Netherlands, reporting a notable yearly incidence increase among Dutch females (APC of 1.6%, 95% CI [1.2–1.9%]) (Al-Jamaei et al., 2022). The rising incidence of OSCC among Indonesian women may represent a new emerging cancer burden, akin to the increase in head and neck cancer among young white women in the US (APC 2.2%) (Patel et al., 2011).

Based on nationwide studies, an estimated 1–2% of Indonesian women (equivalent to 2.3 million individuals) engaged in daily tobacco consumption in 2019. There is growing concern about the successful efforts of the tobacco industry in increasing smoking prevalence among women in recent years (Hardesty et al., 2019). This increasing trend of OSCC cases among Indonesian women observed in our trend analysis may be attributed to changes in lifestyle, including a notable increase in smoking, particularly in urban areas. Tobacco Control data in Indonesia has shown that Indonesian women smoke an average of 1 to 10 cigarettes per day (Barraclough, 1999). National Health Surveys conducted in 2007, 2013, and 2018 also revealed a slight and fluctuating increase in the number of women reporting smoking, approximately 1–2% of the population aged over 10 years old (The Indonesian Ministry of Health, 2008, 2014, 2019).

Notably, in the 5–9 years age group, a higher percentage of females (1.3% (0.8–2.1%)) reported smoking cigarettes daily compared to males (0.9% (0.8–0.9%)) in Indonesia (The Indonesian Ministry of Health, 2019). In Surabaya, the second largest city in Indonesia, women who smoke daily consumed an average of five to six additional cigarettes per day in 2018 compared to previous national surveys in 2011 (Hardesty et al., 2019). Female smokers tend to smoke more cigarettes than male smokers, with women smoking an average of 16 cigarettes per day and men smoking an average of 12 cigarettes per day (The Indonesian Ministry of Health, 2008). A higher risk of head and neck cancer was found among female smokers (odds ratio/OR 2.33, 95% CI [1.56–3.49]) compared to male smokers (OR 1.65, 95% CI [1.14–2.39]) (Hashibe et al., 2007).

Until today, data comparing smoking trends between genders and domiciles (urban/rural areas) in Indonesia are limited. A study in Indonesia found that 24.3% of urban women and 16.5% of rural women had experimented with smoking cigarettes (Koalisi untuk Indonesia Sehat, 2008). Similarly, in the US, the number of Asian women who smoked was higher in urban areas (5.5%) compared to rural areas (1.0%) between 2007 and 2014 (Theilmann et al., 2022). In Greece, smoking prevalence also differed between urban and rural women, with rates of 31.3% and 25.4%, respectively (Gikas et al., 2007). It is worth noting that lifestyle changes like increased smoking were assumed as potential factors contributing to the rising OSCC cases among Indonesian women. A Chinese study supported this notion, showing a decrease in the average age at which urban women began daily smoking across generations, declining from 30.5 years (1930s) to 18.7 years (1990s) (p < 0.001) (Zhang et al., 2022b). Further research explicitly examining gender and urban/rural differences would be valuable.

Limited data exist comparing smoking rates between men and women based on their place of residence. In the US, men’s smoking prevalence was higher (50%), with rural agrarian men having lower rates (41%) than urban men (52%). Overall, women’s smoking rates were lower (24%) than men’s, and the contrast between rural and urban areas was more pronounced (rural agrarian: 10%; urban: 26%), with urban areas having higher smoking rates (Doogan et al., 2017). A Polish study from 2007 to 2010 found non-significant changes in the percentage of current smokers among men and women. More women than men who were former smokers resumed smoking in follow-up (7.3% of women vs. 4.9% of men), and more women who had never smoked at baseline started smoking (0.9% of women vs. 0.4% of men), but these differences were not statistically significant. Sex did not significantly impact the likelihood of being a current smoker, but being a man increased the chances of quitting smoking (OR 1.26, 95% Cl [0.99–1.60] in baseline and OR 1.16, 95% Cl [0.90–1.49] in follow-up). Over the 6 years, the overall percentage of current smokers decreased significantly by 3.1%. More significant declines were observed among men, in urban areas, and among participants born between 1940 and 1960 (Połtyn-Zaradna et al., 2019).

Trend analysis according to tumor sites

Trend analysis according to tumor sites (Table 1) revealed an increasing trend of OSCC originating from the buccal mucosa (APC 8.62%) and the tongue (APC 3.48%). Similar findings were observed in a study conducted in Brazil from 2002 to 2013, but with a lower APC value of 2.4% for cancers of the buccal mucosa (Perea et al., 2018). The study in the Netherlands also supported our findings, demonstrating a significant increase in the incidence of OSCC of the tongue (Al-Jamaei et al., 2022). Smokeless or chewing tobacco (SLT) use, including loose-leaf chewing tobacco, moist snuff, and dry snuff, has been linked to an increased risk of buccal mucosa cancers (Wyss et al., 2016; Khan et al., 2020). The 2014 national survey in Indonesia found that 2.5% of people aged 10 and above were SLT smokers, and 1.6% were occasional tobacco chewers, representing 4.1% of the total smoking population. Regions with the highest proportion of daily tobacco chewers were East Nusa Tenggara (17.7%), West Papua (11.4%), North Maluku (7.1%), Papua (6.7%), and Maluku (5.7%) (The Indonesian Ministry of Health, 2014). Elderly individuals above 75 years had a higher prevalence of SLT use (24.5%) compared to those aged 10–55 years (12%) (The Indonesian Ministry of Health, 2008). Smokeless tobacco may contain carcinogenic agents, such as arsenic, linked to cancer risk (Muthukrishnan & Warnakulasuriya, 2018). A systematic review indicated an increased risk of oral cancer associated with SLT use, with a higher risk ratio in women than men (RR 2.94, 95% CI [2.05–4.20]; p < 0.001) and women (RR 6.39; 95% CI [3.16–12.93]; p < 0.001) (Mu et al., 2021). This aligns with data from the Indonesian National Survey showing that SLT use is more common in women, with a prevalence of 3.9% among men and 4.8% among women (The Indonesian Ministry of Health, 2014).

Other substances and traditions, betel quid (BQ) chewing, common in various Indonesian regions, including Aceh, North Sumatra, Sulawesi, West Kalimantan, and East Indonesia, are associated with an increased risk of buccal mucosa cancer (Wimardhani et al., 2019; Portal Informasi Indonesia, 2019). The BQ package typically includes areca nut, betel (Piper betle) leaf, and slaked lime, sometimes with tobacco (Wimardhani et al., 2019; Portal Informasi Indonesia, 2019). A study discovered that the BQs from different regions in Indonesia contained 92 phytochemical metabolites, with carcinogenic compounds like safrole abundantly present in betel inflorescence from West Papua but not in other regions (Zhang et al., 2022a). Safrole in BQs chewing forms stable safrole-DNA adducts, possibly contributing to oral cancer (Yu et al., 2011).

A pilot study in five Indonesian provinces found that 12.6% of participants (123 out of 973) were BQ chewers. Of these, 88.62% (109 out of 123) chewed BQ without tobacco. Chewing tobacco-free BQ was significantly associated with an increased risk of oral potentially malignant disorders (OPMD) (OR 8.16, 95% CI [5.25–12.68]). BQ chewing remained associated considerably with OPMD even after adjusting for age, gender, alcohol consumption, and cigarette smoking (p < 0.0001) (Maher, Ferreira & Jones, 2023). A case-control study in Jakarta also found that chewing at least one quid per day, especially a combination of betel leaf, areca nut, lime, and tobacco, significantly increased the risk (5–6 fold) of OPMD (Amtha et al., 2014).

These factors, both SLT or chewing tobacco and BQ, may play a role in the observed trend of increasing buccal mucosa in our study. Although not explicitly analyzed, we acknowledge their significance and the need for further investigation. The rising incidence of buccal mucosa cancer is likely influenced by various risk factors, cultural practices, and environmental influences.

Different characteristics between young and old OSCC patients

Our study identified distinct clinicopathological features in young and elderly OSCC patients, particularly in tumor sites and staging. The tongue was a predominant site in our OSCC among young adults, consistent with previous studies (Sasaki et al., 2005; Sharma & Singh, 2016). Identifying specific subsites in both young and old patients is crucial for assessing margin invasion, which is linked to surgical success and patient prognosis (Lee et al., 2022). In our previous study, we observed that tumor subsites, particularly the mouth NOS (OR 3.04; 95% CI [1.17–7.93]) and the palate (OR 6.13, 95% CI [1.73–21.74]), were predictors of margin invasion in the general population. Among young patients, the palate subsite had an exceptionally high association with margin invasion (OR 38.77, 95% CI [3.36–447.66]) along with the presence of LVI had an (OR 11.61, 95% CI [1.34–100.61]), providing valuable predictive indicators (Rahadiani et al., 2022b).

In our present study, only 28.5% of the cases underwent resection, and 12.2% of patients did not undergo staging. Cases with resection specimens encompassed the entire lesion and included all the tumor components necessary for determining tumor size, LN involvement, LVI, and presence of tumor in the resection margins. Among the subjects who underwent tumor resection, the majority had advanced-stage OSCC at the time of diagnosis (81.3%). The prognosis of OSCC worsens with advancing TNM staging (Siriwardena et al., 2020). In Indonesia, young patients aged 45 years or younger showed a higher proportion of advanced-stage OSCC than older patients, which aligns with findings from a report in Brazil (Benevenuto et al., 2012). Their report revealed significant differences in TNM staging between young and old patients, with 66.7% of young patients diagnosed in stages III and IV, compared to 31.2% in older patients (p = 0.039).

Histopathological analysis (Fig. 4 and Tables 2–5) showed that most patients in our study exhibited keratinization, with no significant difference between young and old patients. A lower keratinization degree is associated with a higher recurrence risk and poorer prognosis than those with a higher degree (Wolfer, Elstner & Schultze-Mosgau, 2018). Surface keratinization in the epithelium indicates rapid epithelial maturation, playing a role in maintaining tumor development and promoting differentiation for well-differentiated OSCC in carcinogenesis. In our present study, well-differentiated SCC was the most prevalent histological subtype (53%), while poorly differentiated SCC was the least common across all age groups and clinical stages, aligning with earlier research (Padma et al., 2017).

The relationship between tumor differentiation and prognosis remains controversial, as the WHO histological grading system does not consistently reflect a tumor’s aggressiveness. However, recent research suggests that the level of differentiation can independently predict lymph node metastasis in poorly-differentiated OSCC (Siriwardena et al., 2020). Bryne’s score quantifies the cumulative degree of keratinization, nuclear pleomorphism, lymphoplasmacytic infiltration, and invasion pattern of OSCC (Wagner et al., 2017). Our findings align with studies from the UK (Sasaki et al., 2005) and Thailand (Iamaroon et al., 2004), where grade I tumors were the most prevalent, followed by grade II tumors. Our study found no significant variations in keratinization, differentiation, or Bryne’s scores between young and old patients. However, it is essential to interpret the results with caution. Some patients’ charts were not completely filled out, particularly regarding data on histological differentiation and Bryne’s score of cellular differentiation, which may affect the representativeness of our findings in these aspects.

Different characteristics between female and male OSCC patients

In this present study, although OSCC demonstrated a significant increasing trend in female patients, as presented in Fig. 3 and Table 1, no differences in histopathological tumor characteristics were observed between males and females, as shown in Tables 4 and 5. This study only observed significant sex differences in age and tumor sites, consistent with a prior study in Taiwan (Lin, Hsu & Tsai, 2020). Our study found a greater proportion of female OSCC patients in almost all age groups except for the 41–50- and 61–70-years-old age group. Furthermore, women had a higher proportion of tumors originating from the tongue, lip, and buccal mucosa.

Strengths and limitations

This study has several limitations to be pointed out. Firstly, it was conducted in a single institution, potentially impacting external validity and generalizability. Secondly, discrepancies in assessment by different pathologists could arise due to histologic feature variations of cancer and intratumor heterogeneity. To address this, two independent pathologists were involved in the reassessment to minimize bias. Additionally, the lack of consensus on the most appropriate scoring system for assessing OSCC histological grading remains a subject of debate.

We also acknowledge the potential influence of other factors beyond the scope of our study that may have contributed to the rise in cases of OSCC over the study period. These include increased awareness of health issues, particularly in the oral region, which is highly symptomatic and easily visible (Macpherson, 2018), improved access to healthcare facilities (Aleshin-Guendel et al., 2021), the increased coverage of national insurance programs (Myerson et al., 2020), and advancements in medical imaging diagnostic capabilities (Tancredi, Cullati & Chiolero, 2023). The combined effect of these factors could lead to more frequent hospital visits and improved detection rates, contributing to the observed trends.

Lastly, future cancer incidence projections should be interpreted wisely as uncertain factors can influence them and may not accurately reflect current circumstances. Therefore, it is only amenable to make short-term forecasts (5–10 years), as demonstrated in this study. The predictions for OSCC in the subsequent 5 years (2021–2025) were based on the assumption that they would share the same clinicopathological characteristics observed from 2001–2020. Any dynamic human evolutions, such as plummeting or escalating population size, increasing rates of certain risk factors (e.g., HPV infection or smoking), advancements in diagnosis technology, and improvements in health policies, may influence case forecasting. Additionally, this investigation did not include population-level data, and thus multicentre data should be considered for further evaluation of the mathematical prediction made in this study.

Despite its shortcomings, this epidemiology-based pathological study provides robust data on the prevalence of OSCC in the country. The study involves a relatively large sample and a comprehensive statistical analysis, revealing essential clinicopathological features of OSCC based on age and sex. The 20-year study period was relatively adequate for trend analysis and projected incidence estimation of OSCC for the subsequent 5 years among Indonesian patients. This work is the first publication from Indonesia and provides a valuable contribution to both national and international society, particularly to health authorities. It highlights the increasing trend of oral cancer that should be addressed through policy reform and by raising public awareness of this disease. The results of this study emphasize the importance of oral cancer prevention as an emerging burden in Asia. Future epidemiological and clinical research should focus on identifying risk factors and implementing preventive measures to address the increasing incidence trends.

Conclusions

Based on cancer data from a primary referral hospital, the incidence trend of OSCC in Indonesia showed a moderate increase over the 20-year study period and is projected to increase further in the next 5 years, particularly in women. A significant upward trend was observed in females and in OSCC of the tongue and buccal mucosa. The proportion of OSCC cases among our young patients, most of whom were diagnosed at an advanced stage, was significantly higher than the global average reported in international literature. These findings suggest that focusing research on the carcinogenesis of OSCC in women and young patients, specifically in OSCC of the tongue and buccal mucosa, may help identify important risk factors that could be targeted for preventive measures.

Supplemental Information

Supplemental Information 1 Raw data generated to measure the incidence rate of OSCC in Indonesia’s national tertiary hospital between 2001 and 2020, particularly in young and female patients.

Click here for additional data file.

Supplemental Information 2 Description of Raw Data.

Click here for additional data file.

Supplemental Information 3 STROBE checklist 2007.

Click here for additional data file.

Supplemental Information 4 Trend Analysis using Joinpoint Regression Method.

Comprehensive analysis of the annual incidence trend of oral squamous cell carcinoma (OSCC) using Joinpoint regression analysis among all patients, stratified by age, sex, and tumor origin sites.

Click here for additional data file.

Supplemental Information 5 Best-Fitted Model for Estimated Projection.

In-depth analysis and validation of the generated model used to estimate the projection of oral squamous cell carcinoma (OSCC) cases over the next 5 years, employing linear/non-linear regression analysis.

Click here for additional data file.

Additional Information and Declarations

Competing Interests

Author Contributions

Human Ethics

Data Availability

The authors declare that they have no competing interests.

Nur Rahadiani conceived and designed the experiments, performed the experiments, analyzed the data, prepared figures and/or tables, authored or reviewed drafts of the article, and approved the final draft.

Muhammad Habiburrahman conceived and designed the experiments, performed the experiments, analyzed the data, prepared figures and/or tables, authored or reviewed drafts of the article, and approved the final draft.

Marini Stephanie analyzed the data, authored or reviewed drafts of the article, and approved the final draft.

Diah Rini Handjari analyzed the data, authored or reviewed drafts of the article, and approved the final draft.

Ening Krisnuhoni analyzed the data, authored or reviewed drafts of the article, and approved the final draft.

The following information was supplied relating to ethical approvals (i.e., approving body and any reference numbers):

The Ethics Committee of the Faculty of Medicine Universitas Indonesia and Dr. Cipto Mangunkusumo Hospital have given ethical approval for this research (protocol number 21-03-0246) under the following decision number KET-178/UN2.F1/ETIK/PPM.00.02/2021. Based on our board’s ethical research policy (No: ND-826/UN2.F1/ETIK/PPM.00.02/2022), direct informed consent has been waived for studies using existing data collections, documents, pathological specimens, or other diagnostic specimens.

The following information was supplied regarding data availability:

The raw data are available in the Supplemental Files.

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
