# Peer review of "Estimated projection of oral squamous cell carcinoma annual incidence from twenty years registry data: a retrospective cross-sectional study in Indonesia"

_PeerJ, doi:10.7717/peerj.15911_

## Round 0.1 · original submission · Major Revisions

The authors should pay attention to Reviewer 2 comments, since there are important suggestions to address.

Reviewer 1 ·

Basic reporting

1. In Trend Analysis According to Age section, the authors mentioned their data of “old patients” corroborated with those of US population with tongue cancer from 1973-1997. What was the purpose of this comparison?

Experimental design

No comment.

Validity of the findings

1. Did the authors do any type of validations for their prediction of future cases?
2. The authors stated that “the increasing trend among Indonesian women may be linked to changes in lifestyle among women such as increased smoking for those living in an urban environment.” Was there a significant difference of trend between females living in urban and rural areas? What about males? Was there also an increased trend of smoking?
3. In line 317, “Tobacco Control data in Indonesia demonstrated that Indonesian women smoke an average of 1 to 10 cigarettes per day (Barraclough, 1999).” I assume the data were collected for year 1999? I don’t think tobacco use in one year can suggest smoking as a potential cause of increasing cases in women. What about tobacco use before and among the 20 years? Was there a significant change?

·

Basic reporting

In Tables 2-5, there are columns named '%', where the authors did not specify whether it is row % or column %, which could be misleading, because different calculation of % may lead to different interpretations.

Experimental design

1. The investigation is not rigorous enough. In the section of choosing the optimal line fitting of trend analysis (lines 135-146), the authors proposed to compare the 3 statistics (MAPE, MAD, and MSD) among 3 models (linear, quadratic, and growth). There are several problems with this approach.

Firstly, how the 3 statistics were calculated were not described, i.e., whether they are calculated by fitting the models with all 20 years data and calculate the statistics based on the residuals of fitting, or calculated using cross-validation type of approach (i.e., fitting the models with part of the 20 years data and predict the OSCC cases on the remaining data) should be stated.

Secondly, since the author did not mention how the stats were calculated, intuitively I assume this is calculated based on the former option. In this case, the model choice is less meaningful, because there might be an issue of over-fitting. Linear model is a nested in quadratic model (with the coefficient of t^2 to be 0), thus the fitting of quadratic model must be better than linear model. Therefore, the linear model is very unlikely to be chosen (still possible because better fitting here means less MSD, but not necessarily the other 2, but the 3 statistics are highly correlated so it is very likely quadratic model also have smaller MAPE and MAD than linear model as well), even if linear model is the 'true' model, just because quadratic model is more flexible and may fit 'noise' as signal. Also, for the 'growth' trajectory, it is modelled as y=a*b^t, where there are only 2 parameters, while the quadratic model has 3 parameters, which means the quadratic model is more powerful and thus should fit better in general, but the better fitting might be overfitting.

2. The methods are not stated clearly. For the method of joinpoint regression, there are multiple parameters that users need to specify, including, optimizing method (grid search or Hudson's), maximum join point, and the closeness of different join points. Without these information, it is not clear whether the results are reliable.

3. The goal of separating the trend analysis to joinpoint regression and model selection of 3 different models is not clearly stated. It seems that JP regression was used for fitting the past data while quadratic model was used for prediction the future. However, it is not clear why the authors cannot just use the same model for fitting past data and predicting the future. Specifically, if the model to predicting the future is different from the model for fitting the past trend, what is the point of past data fitting?

Validity of the findings

The results of the trend analysis are not solid and may need further justification.

Firstly, the predicted future cases in 2021-2025 using model selection are not reliable. In all cases, the quadratic model were selected, but it is very likely model over-fitting. Therefore, the predictions does not make too much sense. Meanwhile, as we can see in the fitting plots in data_s3, the residual variances are pretty large, which means that the data is noisy and might be predicted with large variance (regardless of which model is chosen). Therefore, the error bar of predicted numbers should also be provided.

Next, in the trend analysis, the authors needs to check whether the data needs transformation. It seems that larger cases is associated with larger variance. For example, in data S3 part a.1, it seems the residuals for the large y values are larger, which means that they may have a larger influence on the model fitting. Therefore, a logarithm transform of the data might be necessary. I would suggest a box-cox transformation check.

Thirdly, I think an important factor that there are more and more OSCC cases in the recent years is that more and more people and doctors are aware of health problems and they are willing to go to the hospitals. Also, hospitals have more capacities for more patients. These gives more chance of observing OSCC if there are any (with the same diagnostic criterion as used in the paper). 2001-2020 is a long time frame, so this change could not be ignored. I think this might be considered when listing the limitations.

Reviewer 3 ·

Basic reporting

no comment

Experimental design

Method part:
Dear authors, in reviewing histopathologic slides during this 20-year-old, please describe if you had any slides which was excluded because of inappropriate fixation or other improper procedures (exclusion criteria)..?

Validity of the findings

In Discussion Part:
What's the authors opinion about the increasing trend in buccal mucosa? Is there any special type of tobacco use or similar substances in your country..?

Additional comments

no comment

---

## Round 0.2 · accepted · Accept

The authors have addressed reviewers' comments and have properly revised the statistical methods. I am happy with the current form of the manuscript and I really appreciated how the authors performed the revisions.

Reviewer 1 ·

Basic reporting

No comment.

Experimental design

No comment.

Validity of the findings

No comment.